# Integrating digital health technologies into the healthcare system: Challenges and opportunities in Nigeria

Adaeze E. Egwudo[1], Ayodapo O. Jegede[2*], Tolulope A. Oyeniyi[3], Nkolika J. Ezekwelu[1], Samirah N. Abdu-Aguye[4], Azuka P. Okwuraiwe[5], Chizaram A. Onyeaghala[6], Theresa O. Ozoude[7], Muritala O. Suleiman[8], Grace O. Aziken[9], Oluchukwu P. Okeke[10], Olunike R. Abodunrin[11], George U. Eleje[12,13], Folahanmi T. Akinsolu[14], Olajide O. Sobande[10]

1 Department of Community Health and Primary Care, Lagos University Teaching Hospital, Lagos, Nigeria, 2 Department of Clinical Pharmacy and Pharmacy Administration, Obafemi Awolowo University, Ile-Ife, Osun, Nigeria, 3 Department of Public Health and Epidemiology, Nigerian Institute of Medical Research, Lagos, Nigeria, 4 Department of Clinical Pharmacy and Pharmacy Practice, Ahmadu Bello University Zaria, Kaduna, Nigeria, 5 Centre for Human Virology and Genomics, Microbiology Department, Nigerian Institute of Medical Research, Lagos, Nigeria, 6 Department of Internal Medicine, University of Port Harcourt Teaching Hospital, Port Harcourt, Nigeria, 7 Department of Microbiology, Veritas University, Bwari Area Council, FCT Abuja, Nigeria, 8 Department of Human Anatomy, Federal University, Dutse, Jigawa State, Nigeria, 9 Department of Computer Science, University of Benin, Benin City, Edo State, Nigeria, 10 Nigerian Institute of Medical Research Foundation, Yaba, Lagos State, Nigeria, 11 Department of Epidemiology and Biostatistics, Nanjing Medical University, Nanjing, China, 12 Effective Care Research Unit, Department of Obstetrics and Gynaecology, Faculty of Medicine, College of Health Sciences, Nnamdi Azikiwe University, Nnewi Campus, 13 Nnamdi Azikiwe University Teaching, Hospital, NAUTH, Nnewi, Nigeria, 14 Department of Public Health, Lead City University, Ibadan, Nigeria

* dapojegede@oauife.edu.ng

## Abstract

Integrating digital health technologies (DHTs) in Nigeria's healthcare system holds promise, yet the opportunities, challenges, and strategies influencing their success remain insufficiently explored. This scoping review aimed to map these factors, focusing on healthcare settings in Nigeria. A comprehensive search of databases (PubMed, Scopus, Web of Science, and CINAHL) and Google Scholar identified publications on DHT use in Nigeria from July 1, 2014, to June 30, 2024. A total of 31 observational and experimental studies were included involving healthcare workers, patients, caregivers, or other stakeholders impacted by DHT integration. Key findings revealed that DHTs enhanced treatment adherence, healthcare utilization, and community engagement while expanding technology infrastructure for scaling interventions. Notable opportunities included support and training and improved data quality. However, challenges such as operational and logistical barriers, inadequate network coverage, and cultural and gender sensitivity issues were prevalent. Strategies to address these challenges focused on continuous training for healthcare workers, community involvement to foster engagement, and data reporting and quality improvements. Despite their potential to transform healthcare delivery, particularly

**Data availability statement:** All data underlying the findings of this study are fully available and can be accessed without restriction. The data are included as supplementary files with this submission. The extraction template has also been included in the submission.

**Funding:** This work was supported by the Nigerian Institute of Medical Research (NIMR) Foundation [Grant Number NF-GMTP-24-152809]. The funding was received by OOS. The NIMR Foundation website can be accessed at https://nimrfoundation.org. The funders had no role in study design, data collection and analysis, decision to publish, or preparation of the manuscript.

**Competing interests:** The authors have declared that no competing interests exist.

in underserved areas, successful integration of DHTs in Nigeria requires addressing infrastructure gaps, cultural norms, and operational challenges. Community engagement, capacity building for healthcare workers, and data-driven decision-making are critical to maximizing the impact of digital health interventions in Nigeria.

## Author summary

This scoping review provides a comprehensive overview of the published literature on integrating digital health technologies (DHTs) into healthcare settings in Nigeria. These tools, including mobile health applications and electronic health records, have demonstrated the potential to improve patient care, increase access to healthcare services, and enhance data collection for informed decision-making. Our findings indicate that DHTs have contributed to better health outcomes, improved healthcare delivery in underserved and remote areas, and strengthened training for healthcare workers. Despite these benefits, several challenges hinder effective implementation. These include poor internet connectivity, financial constraints, and the need for culturally sensitive program designs. Successful implementation strategies identified in the review involved engaging local leaders and healthcare workers through community involvement, providing tailored training, and improving infrastructure, such as enhancing internet access and ensuring the availability of digital devices. This review underscores the transformative potential of digital health tools in low-resource settings like Nigeria. It also highlights actionable strategies to overcome barriers, offering valuable insights for healthcare providers, policymakers, and researchers working to optimize the use of digital health technologies in similar contexts.

## Introduction

Digital health technologies (DHTs) encompass tools and services that utilize information and communication technologies (ICTs) to optimize health management and improve the prevention, diagnosis, and treatment of diseases [1]. DHTs are broadly classified into telemedicine, mobile health (mHealth), health information technology, wearables, and personalized medicine [2], which hold significant promise for enhancing healthcare delivery, particularly in regions like Africa, where healthcare systems face unique challenges, such as shortage of healthcare providers, poor infrastructure, and the high burden of infectious and non-communicable diseases [3].

The scope of applications of Digital health tools/technologies in Africa has expanded to include health system strengthening, maternal and child health, oncology, and chronic disease management [4–7]. These technologies also improve healthcare access and quality by expanding mHealth solutions that facilitate remote consultations and health information dissemination [8]. There has also been an increase in the adoption of telemedicine, particularly in response to the COVID-19 pandemic [9]. Furthermore, electronic health information technologies have improved

healthcare quality by digitizing patient data, enhancing care continuity, reducing medical errors, and supporting efficient decision-making [10].

Nigeria, the most populous country in Africa, struggles with high healthcare costs and the burden of infectious and non-communicable diseases [11]. With a doctor-to-population ratio of approximately 1:9,083 [12], the country falls well below the World Health Organization's recommended minimum threshold of 2.3 skilled health professionals (including doctors, nurses, and midwives) per 1,000 population required to deliver essential health services. [13]. This shortfall, continues to limit access to quality healthcare acros the country. The adoption of digital health technologies may help resolve some of these challenges. It is, therefore, essential to understand the current extent of the use of DHTs in Nigeria and the barriers and facilitators responsible for its integration into the healthcare system.

This study's conceptual framework combines the Digital Health Ecosystem framework [14] and the WHO Health System Strengthening Model [15] to understand the role of digital tools in healthcare. The Digital Health Ecosystem highlights the interconnectedness between digital tools (e.g., telemedicine, mobile apps) and healthcare stakeholders (e.g., providers and patients), emphasizing how these systems work together to improve patient care and health outcomes [14]. The WHO model emphasizes strengthening health system components like service delivery, workforce, and information systems [15]. These elements interact and can be used as a framework to identify operational challenges and opportunities.

This scoping review mapped the challenges and opportunities to maximize the benefits of integrating digital health technologies into healthcare systems and proposed ways to improve the health system efficiency in Nigeria.

## Methods

This was a scoping review conducted to explore opportunities and challenges associated with integrating DHTs into healthcare systems in Nigeria. The review followed the Arksey and O'Malley framework [16] and adhered to the PRISMA checklist for scoping reviews. (See S1 Appendix) The review was registered with Open Science Framework (DOI: https://doi.org/10.17605/OSF.IO/D3J9S) and reported using the PRISMA extension for scoping reviews [17–18].

### Research questions

The research questions guiding the review included:

1. What are the key challenges hindering DHT integration into Nigeria's healthcare systems?

2. What strategies have been used successfully in integrating digital health technologies in Nigeria, and what lessons have been learned from these approaches?

### Search strategy

A systematic search was conducted on PubMed, Scopus, Web of Science, CINAHL, and Google Scholar to identify relevant literature published within the last ten years (Jul 1, 2014 - Jun 30, 2024) to ensure the findings reflected current trends and challenges in DHT integration. The search was conducted using the keywords and Boolean operators "Digital health technologies" AND "Integration" AND "Healthcare systems" AND "Sub-Saharan Africa". The searches were limited to articles published in English, with the most recent search executed on Aug 28, 2024. Although the study focused on Nigeria, the keyword Sub-Saharan Africa was used to determine what had been done within the region before honing down on Nigeria. (S2 Appendix).

### Inclusion and exclusion criteria

Observational and experimental studies (qualitative, quantitative, and mixed-method studies) that focused on healthcare workers, patients, caregivers, policymakers, and technology developers who were involved in or affected by DHT

integration were included in the scoping review. In addition, we included studies that focused on the Nigerian healthcare system and addressed strategies, challenges, and opportunities for integrating DHTs into the healthcare system. Systematic reviews, scoping reviews, reports, editorials, opinion pieces, conference abstracts, and non-peer-reviewed articles were excluded from the review. Additional exclusions were made for studies that did not specify a digital health technology [19,20], lacked a defined study population for the intervention [21], employed an unsuitable study design [22,23], or were protocol-only publications [24].

## Study selection

A two-phase selection process was conducted on the titles and abstracts retrieved from the electronic search, using Rayyan, removing duplicate articles to ensure a rigorous and unbiased inclusion of studies.

The *first phase* involved screening the titles and abstracts of the articles based on the eligibility criteria. Two reviewers (N.J.E. and S.A.A.) independently did this to ensure unbiased selection, with discrepancies resolved through discussion and by a third reviewer (A.O.J.). Full text of studies that passed the initial screening were retrieved for full-text review. The *second phase* was a full-text screening of the included articles by three authors (N.J.E., S.A.A., and A.O.J.) to ensure they met the predefined inclusion criteria. The three authors did the full-text review independently, and discrepancies were resolved by either A.E.E. or T.A.O.

The review excluded 11 articles (detailed in S3 Appendix) based on specific criteria aligned with the study's focus on implemented DHTs. The reasons for exclusion varied, including studies that lacked a defined DHT intervention, focused only on knowledge or perceptions without implementation or were preliminary or protocol-based studies.

## Data extraction

A data extraction sheet was developed and pre-tested with five studies to ensure consistency and effectiveness before full-scale extraction. Three authors independently extracted data (N.J.E., S.A.A, and A.O.J.). The extracted data included study characteristics (e.g., author, year, location, sample size), study population (e.g., healthcare workers, patients, caregivers), research methods (e.g., qualitative, quantitative, mixed), type of digital health technology (e.g., mHealth, Electronic Health Records (EHRs), telemedicine), opportunities and challenges, and strategies used to overcome challenges. (See S4 Appendix).

## Methodological quality and risk of bias assessment

The methodological quality of the included studies was appraised using the Mixed Method Appraisal Tool [25]. Two independent reviewers (A.E.E. and T.A.O.) conducted the quality assessment, resolving discrepancies by a third reviewer (S.A.A.). Total scores for each study type were calculated and ranged from 0 to 100%. (S5 Appendix)

## Synthesis of findings and presentation

Descriptive analysis was used to summarise the characteristics of the included studies, while tables and figures were used to organize extracted information about the study's characteristics and key findings. Key data points, such as types of DHTs, outcomes, challenges and opportunities, were extracted and analyzed using thematic analysis to identify common challenges, opportunities, and solutions across studies.

Table 1 shows the conceptual framework developed for the study of the use of digital health technology for integrated health systems in Nigeria based on the outcomes of this study.

The study's conceptual framework blends elements from the proposed Digital Health Ecosystem Framework (DHEM) [14] and the WHO Health System Strengthening (HSS) Model [15], creating a comprehensive approach to understanding the integration of digital health technologies in healthcare settings. These elements include service delivery, health

**Table 1. Study themes & sub-themes based on the study's conceptual framework.**

| Main Theme | Sub-Themes | Link to Frameworks |
|---|---|---|
| Service Delivery | Health outcomes, Operational & logistics issues | Tied to "Service Delivery" (HSS) and process improvements (DHEM). |
| Health Workforce | Digital literacy, support & training | Linked to "Health Workforce" (HSS) and people-centred readiness (DHEM). |
| Information System | Data reporting, Learning Opportunities | Combines "Information Systems" (HSS) with the DHEM focus on digital integration. |
| Financing | Cost analysis, Resource constraints | Addresses "Financing" (HSS) with contextual constraints emphasized in DHEM. |
| Leadership & Governance | Real-time data, Decision-making | Reflects "Governance" (HSS) and decision-support technologies in DHEM. |
| Technology Infrastructure | Technology infrastructure | Aligns with the "Infrastructure" layer of DHEM and "Access to Essential Medicines" (HSS). |
| Stakeholder Engagement | Community engagement & support | Stresses engagement (DHEM) within "Service Delivery" and "Governance" (HSS). |
| Cultural & Social Factors | Cultural sensitivity | Introduces cultural dimensions affecting "Service Delivery" (HSS) and inclusivity in DHEM. |
| Policy & Regulation | Security | Encompasses regulatory measures from both frameworks. |
| Sustainability & Scalability | Scalability, Network coverage | Relates to the DHEM emphasis on long-term digital health initiatives and their integration into HSS. |

workforce, and technology infrastructure, which influence how digital health is adopted and sustained, interacting to address the operational challenges and opportunities within healthcare systems.

## Results

### Selection of studies

A total of 4790 records were retrieved, one thousand three hundred twenty-eight duplicates were removed, 3,418 were excluded after the title and abstract screening, and 11 studies were excluded after full-text review. The PRISMA fl\ow chart is presented in Fig 1. The details of the 31 studies [26–56] that met the eligibility criteria are reported in Table 2.

### Characteristics of included studies

The included studies were conducted between 2016 and 2024, and the total sample size for the studies was 122,572, with mean age ranging from 20-45 years (Table 2). Nine (29%) of the 31 studies were conducted in Lagos, Oyo, Ondo and Osun states located in the southwestern geopolitical region of Nigeria [26,29,34,36,39,44–46,48], and three (9.7%) were nationwide studies [43,51,52]. Twenty-four (77.4%) studies were conducted within facility settings [26,27,30,33–41,44–52,54–56]. The included study participants included healthcare workers [27,29,30,33–36,38,39,41–43,46,49,51–54], patients [26,27,33,37,39–41,44,45,47–50,53,55,56], and community-dwelling individuals [28,31,32,34,42,54].

The study research methodologies include eight (27%) qualitative studies [35,37–39,42,45,55,56], seven (23.3%) mixed method studies [4,5,8,9,11,29,35], and nine (29.7%) quantitative studies (quasi-experimental and randomized controlled trials) [27,40,43,44,49,54]. Nineteen (61.3%) studies were interventions on maternal, newborn, and child health (MNCH), which include immunization [28,29,32,34–36,42,51,52,54–56], antenatal and postnatal care [40,44,45,48], family planning [31], and child health [38,41]. Other studies were on depression [26], tuberculosis [27], hypertension [33,43], and HIV [39].

As reflected in Table 2, the majority of studies focused primarily on user-level populations. Healthcare workers (HCWs) were the most frequently studied group, appearing in eighteen out of the thirty-one studies reviewed (58.1%)

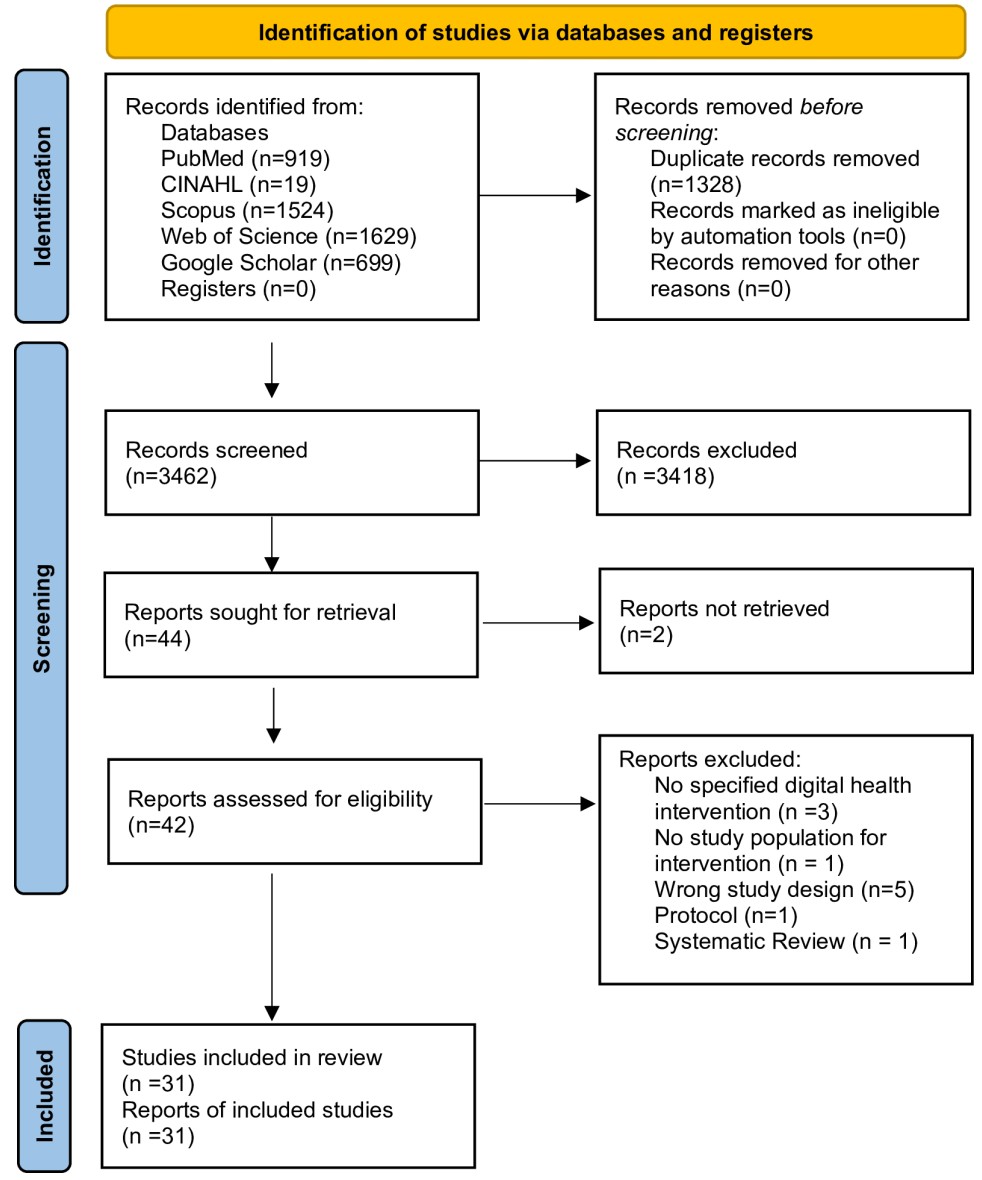

**Fig 1. PRISMA flow diagram for search results.**

[27,29,30,33–36,38,39,41–43,46,49,51–54]. Patients or caregivers were featured in seventeen studies (54.8%), reflecting significant attention to end-user experiences with digital health technologies [26,27,33,34,37,39–41,44,45,47–50,53,55,56]. Community members, adolescents, or members of the general public were included in five studies (16.1%) [28,31,32,42,54], often in relation to community-level engagement or public health programs. Only one study explicitly included policymakers as part of the study population [34], and notably, none of the reviewed studies directly examined local digital health technology producers, service providers, or private sector implementers.

Table 2 also identified a significant issue, the short-term, one-off nature of many DHT integration studies, which often lacked continuity or plans for scale-up. For instance, eleven of the studies were pilot projects with limited follow-up [26,28,31,32,37,41,43,46,49,53,54], while nine others were structured as feedback or assessment studies without

**Table 2. Characteristics of included studies.**

| S/N | Author | State(s) | Study setting | Study population | Study design | Condition(s) being studied | Sample size (Male, Female) | Mean age (Years) | Study Type | Quality Assessment |
|---|---|---|---|---|---|---|---|---|---|---|
| 1 | Adewuya et al., 2019 [19] | Lagos | Facility based | Patients | Experimental | Depression | 895 (M=399; F=496) | 34.9 | Pilot | 80% |
| 2 | Akamike et al., 2021 [20] | Ebonyi | Facility based | HCWs, Patients | Experimental | TB | 285 (M=89; F=196) | HCWs: (40.43 Intervention; 33.15 Control) Patients: (38.87 Intervention; 40.02 Control) | Scale-up | 40% |
| 3 | Akande et al. 2024 [21] | Kwara | Community-based | Community members (In-school adolescents) | Experimental | Sexual and reproductive health | 1280 | Not specified | Pilot/ Feedback | 100% |
| 4 | Akeju et al.,2022 [22] | Ondo | Community-based | HCWs | Experimental | MNCH | 24 (M=8; F=16) | Not specified | Feedback | 80% |
| 5 | Ayamolowo et al2023. [23] | Ogun | Facility based | HCWs (Nurses) | Experimental | Not specified | 240 | Not specified | Feedback | 100% |
| 6 | Babalola et al.,2019 [24] | Kaduna | Community-based | Community individuals | Experimental | Family planning | 565 (F =565) (221 intervention and 344 control) | 26.8 | Pilot | 40% |
| 7 | Birukila et al.,2016 [25] | Kaduna | Community-based | Community individuals | Observational | Polio immunization | 21242 | Not specified | Pilot | 100% |
| 8 | Cremers et al.2018 [26] | Lagos | Community and Facility-based | Patients, HCWs | Experimental | Hypertension | 373 | Not specified | Need Assessment | 100% |
| 9 | Ebenso et al. 2021 [27] | Ondo, Kano, FCT | Community and Facility-based | HCWs, pregnant women and policymakers | Experimental | MNCH | 294 | Not specified | Assessment/ Feedback | 100% |
| 10 | Fox et al., 2022 [28] | Enugu | Facility based | HCWs | Experimental | MNCH | 20 | Not specified | Test | 100% |
| 11 | Hicks et al.2021 [29] | Ondo, Kano, FCT | Community and Facility-based | HCWs | Experimental | MNCH | 328 (M=108; F=220) | Not specified | Assessment/ Feedback | 100% |
| 12 | Itanyi et al., 2023 [30] | Benue | Facility based | Patients | Experimental | Not specified | 35 (F=35) | 24 | Pilot | 100% |
| 13 | Kenny et al2017 [31] | Enugu | Facility based | HCWs | Experimental | Child health | 20 | Not specified | Feedback | 10% |
| 14 | Kuhns et al., 2021 [32] | Oyo | Facility based | Patients, HCWs | Experimental | HIV | 27 (M=16; F=11) | 20 | Test | 100% |
| 15 | McNabb et al., 2015 [33] | Abuja, Nasarawa | Facility based | Patients | Experimental | ANC services | 267 (F=267) | 24 | Test | 80% |

*(Continued)*

Table 2. (Continued)

| S/N | Author | State(s) | Study setting | Study population | Study design | Condition(s) being studied | Sample size (Male, Female) | Mean age (Years) | Study Type | Quality Assessment |
|---|---|---|---|---|---|---|---|---|---|---|
| 16 | Musa et al., 2023 [34] | Kaduna | Facility based | HCWs, Patients | Experimental | Child health | Not specified | Not specified | Pilot | 0% |
| 17 | Obi-Jeff et al., 2022 [35] | Kebbi | Community based | HCWs, Community members | Experimental | Routine Immunization | 144 (M=99; F=45) | 39 | Test | 100% |
| 18 | Odu et al., 2024 [36] | Nationwide | Community-based | HCWs | Experimental | Hypertension | 431 | Not specified | Pilot | 60% |
| 19 | Olajubu et al., 2020 [37] | Osun | Facility based | Patients | Experimental | Postnatal Care | 380 (F=380) | 29.54 (intervention), 27.75 (control) | Test | 60% |
| 20 | Olajubu et al., 2022 [38] | Osun | Facility based | Patients | Experimental | Postnatal Care | 20 (F=20) | Not specified | Test | 100% |
| 21 | Olayiwola et al., 2020 [39] | Lagos, Ibadan, Ogun, | Facility based | HCWs | Observational | Not specified | 10 | Not specified | Pilot | 10% |
| 22 | Onyeabor et al. 2024 [40] | Enugu | Facility based | Patients | Experimental | Not specified | 20 (M=11; F=8) | Not specified | Feedback | 100% |
| 23 | Osanyin et al., 2022 [41] | Lagos | Facility based | Patients | Experimental | ANC | 458 (F=458) | Not specified | Test | 60% |
| 24 | Otu et al, 2022 [42] | Cross River | Facility based | HCWs, Patients | Experimental | Non-communicable diseases | 24 (M=2; F=21) | Not specified | Pilot | 80% |
| 25 | Schmitz et al. 2022 [43] | Adamawa | Facility based | Patients (Children) | Observational | Not specified | 1929 (M=955; F=939) | 24-59 | Test | 60% |
| 26 | Shuaib et al. 2019 [44] | Kano, Country-wide | Facility based | HCWs | Observational | Routine Immunisation (RI) | Not specified | Not specified | Pilot/Scale-up | 80% |
| 27 | Tegegne et al. 2018 [45] | Country-wide | Facility based | HCWs | Observational | Routine Immunization | 90396 | Not specified | Test | 100% |
| 28 | Tripathi et al. 2020 [46] | Ebonyi, Kaduna | Community-based | HCWs, Patients, or caregivers | Experimental | Fistula diagnosis and treatment | Not specified | Not specified | Pilot | 100% |
| 29 | Uba et al. 2021 [47] | Kano | Community and Facility-based | HCWs, Community members | Experimental | Routine Immunization | 2735 | Not specified | Pilot | 80% |
| 30 | Udenigwe et al. 2022 [48] | Edo | Community and Facility-based | Patients | Experimental | Maternal Healthcare | 66 (M=27; F=39) | Men- 45 Women- 26 | Feedback | 100% |
| 31 | Udenigwe et al. 2022 [49] | Edo | Community and Facility-based | Patients | Experimental | Maternal Healthcare | 64 (M=25; F=39) | Men- 45 Women- 26 | Feedback | 100% |

HCW: Healthcare workers, TB: Tuberculosis, MNCH: Maternal, newborn and child health, ANC: Antenatal care,

long-term implementation strategies [29,30,33,34,36,38,47,55,56]. This limited design approach undermines the sustainability and long-term impact of digital interventions. Furthermore, the geographic spread of these studies was often narrow, typically focused on one or two states [26–33,35,37–42,44,45,47–50,53–56], which restricts the generalizability and scalability of the findings.

## Opportunities for the implementation of digital health in Nigeria

Table 3 identified several opportunities for implementing DHTs in Nigeria. Health outcomes improved significantly, including better treatment adherence, higher recovery rates, and increased knowledge on pregnancy, postnatal care, and HIV testing [26–28,31,35,37,39,40,42,44–47,50,56]. mHealth interventions expanded healthcare access in remote areas [28,30,41,43,44,46,49,51,54], while telemedicine, electronic health records, and pharmacy-based care improved diagnostics and service reach. Digital literacy and mobile-based educational programs enhanced care outcomes, especially for maternal health [33,40,44,48]. Community engagement, local leader involvement, and gender-sensitive approaches boosted mHealth program effectiveness [32,37–39,42,49,50,53,55,56]. Increased smartphone use supported healthcare expansion [28,36,41,47,48], while digital platforms improved clinical skills and data accuracy, enabling real-time data collection, disease surveillance, and cost-effective healthcare [34,35,37,51,54].

## Challenges to the implementation of digital health in Nigeria

Table 3 highlights the challenges associated with implementing digital health tools in Nigeria, as reported across various studies. Three studies noted a limited impact on risky sexual behaviours and adherence to preventive treatment guidelines [27,28,50]. Large-scale mHealth interventions for sexual and reproductive health encountered significant barriers, including issues of equity and access [28], negative perceptions [35], and contextual constraints [28,31,34,40,41,51,52,56]. Outcomes were further worsened by factors such as the COVID-19 pandemic, high staff turnover, and prolonged consultation times [28,31,34].

Technical challenges were prevalent, with inadequate technology, poor phone battery life, and weak network coverage cited as significant issues [35,39,42,47]. Disruptions to services were also frequently reported [35,36,44–46,49]. Additionally, digital literacy gaps among healthcare workers and participants [30,31,37,42,48] hindered the effective use of digital tools, alongside fears of scams and limited user engagement [42].

Cost constraints [29,37,45,53,55,56], logistical difficulties [36,45,48,52,53], and gender disparities in phone ownership [45,53,55] further restricted access. Privacy and confidentiality concerns emphasized the urgent need for secure digital infrastructure to support implementation [37,39,50,54].

## Strategies used in mitigating challenges to the implementation of digital health in Nigeria

Table 3 outlines various strategies to address challenges in implementing digital health technologies in Nigeria. The use of surveys and interviews improved data quality [30] while tracking the impact of digital tools over time [33–34] and utilizing digital dashboards [54] enhanced healthcare delivery. Video-based training effectively boosted healthcare workers' confidence in maternal and child health services [36], and mHealth training significantly improved outcomes for managing non-communicable diseases [49]. Culturally sensitive interventions were particularly impactful in supporting marginalized groups [53].

Community engagement played a central role in the success of digital health initiatives. Strategies such as text messages, health dramas, and the involvement of local leaders fostered acceptance and usage of digital health tools [26–28,31,35,37,45,47,49,51,55,56]. Reliable infrastructure [36,41,42,44,46,55], financial support [51,54,56], and targeted training for healthcare workers [27,29,35,38–41,43,49–52,54,56]were critical enablers of successful implementation.

Real-time data facilitated informed decision-making [29,40,51,52], while culturally respectful approaches improved health outcomes [53,55,56]. SMS-based education empowered patients to make informed healthcare decisions [37,45],

**Table 3. Thematic analyses of the opportunities, challenges and strategies of the digital health technologies reported in the included studies.**

| S/No | First Author | Opportunities | Challenges | Strategies |
|---|---|---|---|---|
| 1 | Adewuya et al., 2019 [19] | **Health outcomes-** Improved treatment adherence, higher recovery rates, better quality of life and better retention in treatment. | – | **Community engagement & support-** Use of text messages to reinforce adherence, remind patients to attend clinic appointments and monitor disease progression. |
| 2 | Akamike et al., 2021 [20] | **Health Outcomes-** Improved knowledge and adherence to the Isoniazid Preventive Therapy (IPT) guidelines among health workers. | **Health Outcomes-** Adherence to IPT Guidelines Varied by Hospital Type. | **Support & Training-** Regular on-site visits; **Community Engagement & Support-** follow-up calls, frequent reminders. |
| 3 | Akande et al. 2024 [21] | **Health Outcomes-** Improvement in sexual and reproductive health; **Technology- Infrastructure** feasibility of mHealth interventions; **Scalability & Population Reach-** potential large-scale implementation. | **Health Outcomes-** No significant reduction in risky sexual behaviour; **Scalability & Population reach-** access and equity issues; **Contextual & Resource Constraints-** disruption due to COVID-19. | **Community Engagement & Support-** Engagement with mHealth curriculum and targeted interventions for adolescents; adolescents were randomized into intervention and control groups; data on knowledge attitude and practice collected before and after, immediately and after intervention. |
| 4 | Akeju et al., 2022 [22] | **Support & Training-** Increased HCW motivation and satisfaction, increased confidence to carry out clinical roles and improvements in the standard of care provided. | **Cost & Financial access-** lack of continuity of the interventions. | **Support & training-** The VTR Mobile Application allowed HCW participants to access video, audio, and text-based MCH materials through the Internet; **Real-time data & decision-making-** A Data Digitization Application that was a computer-enabled decision support and point-of-care data capture solution that enabled HWs to capture and transmit patient-level health information. |
| 5 | Ayamolowo et al. 2023 [23] | **Technology Infrastructure-** improved training and infrastructure, functional EHR-related infrastructure, sociodemographic factor. | **Digital literacy & education-** brain drain and increased workload, complexity and terminology issues. | **Data Reporting & Quality-** purposive explanatory mixed method study; quantitative study through semi-structured questionnaire; qualitative data through in-depth interview. |
| 6 | Babalola et al., 2019 [24] | **Health outcomes-** increased uptake of health services and health knowledge. | **Digital literacy & education-** low numeracy levels of participants, **Contextual and Resource Constraints-** Participant attrition and lag time between recruitment and start of the intervention, length and numbers of calls. | **Community engagement & support-** The Smart Client digital health tool used fictional role models who demonstrated the desired behaviours and behaviour change process in a drama format. This approach allows the intended audience to observe an action, understand its consequences, and become motivated to repeat and adopt it. |
| 7 | Birukila et al., 2016 [25] | **Community engagement and support-** Provision of basic health information on immunization and engaging with community members about vaccines. | – | **Scalability & population reach-** Use of Bluetooth to rapidly provide accurate health messages to a large population with low literacy and poor socio-economic status. |
| 8 | Cremers et al. 2018 [26] | **Technology Infrastructure-** Feasibility of pharmacy-based care, enhanced care model; **Digital literacy & education-** patient understanding. | **Scalability & Population Reach-** Implementation barriers. | **Data Reporting & Quality-** Observation, focus group discussions, in-depth interviews. |

*(Continued)*

**Table 3.** (Continued)

| S/No | First Author | Opportunities | Challenges | Strategies |
|------|--------------|---------------|------------|------------|
| 9 | Ebenso et al. 2021 [27] | **Technology Infrastructure Learning Opportunities-** Improved clinical skills through the use of clinical videos; **Support & Training-** better-trained health workers in service delivery; **Data Reporting & Quality; Real-Time Data & Decision Making-** enhanced information and data management for policy decisions and governance. | **Contextual & Resource Constraints-** increased workload, poor Internet, inconsistent electricity supply; **Scalability & Population Reach-** sustenance of the project. | **Data Reporting & Quality-** Data collection combined documents review with 294 semi-structured interviews of stakeholders across four phases (baseline, midline, endline, and 12-months post-project closedown) to assess acceptability and impacts of digital interventions. |
| 10 | Fox et al., 2022 [28] | **Support & Training-** Uniform assessment of patients; **Data Reporting & Quality-** creating reliable records; **Health outcomes-** reducing stress, Improving guardian confidence; **Cost & Financial Access-** Reducing costs of treatment. | **Network coverage, technology infrastructure-** Network issues, battery life and lack of electricity. | **Community Engagement & Support-** Pre-implementation engagement using presentations and FGDs; **Support & training-** Training and support of PHC workers and community education. |
| 11 | Hicks et al. 2021 [29] | **Scalability & Population Reach-** Improved knowledge and practices of the e-learning video-based intervention, enhanced acceptance and usage; **Support & training-** Empowerment of FHWs by increasing their confidence and competence. | **Network Coverage & Power-** (Poor internet connectivity, electricity; **Operational & Logistical Challenges** Supply issues and organizational workload. | **Technology Infrastructure-** A VTR mobile intervention was delivered to FHWs in the 3 states. **Health Outcomes-** Changes in workers' knowledge and confidence in delivering MNCH services were examined through a pre-and post-test survey. Stakeholders' experiences with the intervention were explored through semi-structured interviews. |
| 12 | Itanyi et al., 2023 [30] | **Data reporting & quality-** Improved record-keeping; **Health outcomes-** Enhanced maternal newborn and child health services; **Community engagement & support-** Increased communication and positive influence of family support systems. | **Cost & Financial access-** Cost associated with smart cards. **Digital literacy & education-** Difficulty understanding how to use the smart cards; **Security concerns-** Unauthorized use leading to issues on confidentiality and privacy. | **Learning Opportunities-** Consistent messaging to explain the benefits of using smart cards. **Community engagement & support-** Incentives like free health tests; **Support & training-** training of health workers to ask security questions before releasing any health information and availability of healthcare workers and research assistants to provide information about the cards. |
| 13 | Kenny et al. 2017 [31] | **Scalability & Population Reach-** To extend mHealth projects beyond the pilot stage); **Community Engagement & Support-** Encourage positive predisposition to mHealth interventions among primary end users | **Data Reporting & Quality-** Small sample size and sample data. | **Support & Training-** Engaged PHC workers involved in healthcare delivery via interview, presentation and focus group discussion. |
| 14 | Kuhns et al., 2021 [32] | **Health Outcomes-** Promotion of HIV testing and treatment. **Community Engagement & Support-** Engagement using widespread use of text messages and social media. | **Technology Infrastructure-** Lack of airtime on mobile phones, cracked screens on mobile phones that limit the ability to read SMS, loss or theft of phones; **Security Concerns-** Concerns around privacy/confidentiality of the SMS. | **Training & Support-** Training and support for volunteers; **Security Concerns-** promoting confidentiality. |

*(Continued)*

**Table 3.** (Continued)

| S/No | First Author | Opportunities | Challenges | Strategies |
|---|---|---|---|---|
| 15 | McNabb et al., 2015 [33] | **Health Outcomes-** The m4Change intervention showed significant potential to enhance the quality of antenatal care and increase overall client satisfaction; **Digital Literacy & Education-** The intervention facilitated improved counselling and education of health workers, leading to better health outcomes for clients. Furthermore, using standardized health education messages through the mobile application can influence maternal health behaviours positively. **Cost & Financial Access-** Audio-recorded counselling messages are a low-cost strategy for ensuring clients are exposed to standardized health education related to key pregnancy and childbirth issues. | **Contextual & Resource Constraints-** Inadequate commodities management and staffing. Additionally, the unavailability of essential supplies, like tetanus toxoid injections, hindered the effectiveness of services provided through digital solutions. High baseline satisfaction scores also created a ceiling effect, making it difficult to measure improvements in client satisfaction accurately. | **Support & Training-** The m4Change mobile ANC application was implemented to guide CHEWs through patient encounters, enhancing both technical and counselling aspects of care; **Digital Literacy & Education-** Standardized counselling messages were introduced to ensure clients received consistent health education related to pregnancy and childbirth. **Real-Time Data & Decision Making-** The study emphasized the importance of evaluating digital health interventions within real-world conditions to understand their effectiveness in improving service quality without additional program inputs. |
| 16 | Musa et al., 2023 [34] | **Technology Infrastructure-** Expand access to specialized care and leverage increasing mobile and internet penetration to enhance healthcare delivery, especially in remote areas. | **Contextual & Resource Constraints-** Infrastructure deficits, lack of skilled personnel, and absence of clear regulations governing telemedicine in Nigeria. | **Community Engagement & Support-** Collaboration with stakeholders; **Technology Infrastructure** (improve infrastructure); **Support & Training-** Training healthcare personnel to integrate telemedicine effectively into the system. |
| 17 | Obi-Jeff et al., 2022 [35] | **Health Outcomes-** Increased demand and uptake of immunization services; **Community Engagement &Support-** Community-level engagement contributing to acceptability and effectiveness. | **Data Reporting & Quality-** Network issues; **Technology Infrastructure-** Lack of access to and inconsistent use of mobile phones; **Digital Literacy & Education-** low literacy level (inability to read text messages or access the message), fear of scams (ignore messages). | **Technology Infrastructure-** Use of proxy phones; **Digital Literacy & Education-** Localized SMS in the Hausa language. |
| 18 | Odu et al., 2024 [36] | **Support & Training, Learning Opportunities-** Effective knowledge delivery, broad accessibility and cost-effective training of PHC-based HWs in Nigeria on best clinical practices in hypertension management. | **Learning Opportunities-** Variable knowledge gains amongst cadre, limited participation among doctors, short-term evaluation, and rushed learning. | **Support & Training-** Training course endorsed by the federal ministry of health, tailored course content leveraging on existing programs. |
| 19 | Olajubu et al., 2020 [37] | **Digital Literacy & Education; Health Outcomes-** The mHealth intervention in appointment reminders and educational text messages significantly impacted the spectrum of postnatal care utilization outcome measures among the respondents. The intervention also provided educational content emphasizing routine postnatal care's importance, potentially enhancing maternal health outcomes. | **Operational & Logistics Challenges-** Participants lost to follow-up; **Network Coverage & Power-** Erratic power supply occasionally delayed the SMS delivery to some respondents whose phones were off due to a flat battery. | **Technology Infrastructure-** A postnatal care assistant software was developed to send text messages to the phone numbers of the mothers; **Digital Literacy & Education** The SMS provided educational messages about maternal health and reminders about postnatal care attendance. |

*(Continued)*

**Table 3.** (Continued)

| S/No | First Author | Opportunities | Challenges | Strategies |
|---|---|---|---|---|
| 20 | Olajubu et al., 2022 [38] | **Health Outcomes-** Improved knowledge on pregnancy, childbirth and postnatal care, increased utilization of postnatal care services and positive behavioural change. | **Cost & Financial Access-** Financial barriers; **Operational & Logistics Challenges-** Time constraints and technology issues; **Cultural & Gender Sensitivity-** Cultural influence. | **Learning Opportunities/ Community Engagement & Support-** The study used a mHealth approach where educational and reminder messages were sent to pregnant women and their partners via SMS. |
| 21 | Olayiwola et al., 2020 [39] | **Technology Infrastructure-** eConsults improved healthcare access for underserved populations and provided educational benefits to GPs. | **Network Coverage-** Limited broadband connectivity in Nigeria caused delays and frustrations for GPs, significantly increasing case creation times. | **Technology Infrastructure-** The RubiconMD platform was implemented to facilitate eConsult, and feedback tools like rating systems were used to assess its impact and improve healthcare delivery. |
| 22 | Onyeabor et al. 2024 [40] | **Health Outcomes-** Enhanced remote care management, improved usability, structured telemedicine approach); **Scalability & Population Reach-** Potential for widespread adoption. | **Technology Infrastructure-** Technology and infrastructural problem, subjective evaluation, inability to include real patients. | **Community Engagement & Support** Qualitative and quantitative research methods used for evaluating web-based remote patient monitoring system; 20 patient samples randomly selected to evaluate usability and user experience based on system usability scale. |
| 23 | Osanyin et al., 2022 [41] | **Digital Literacy & Education, Scalability & Population Reach-** Expanded scalable digital health solutions by optimizing intervention designs and taking advantage of increasing smartphone penetration in Nigeria. | **Digital Literacy & Education-** Overcome barriers like low education levels); **Operational & Logistics Challenges-** Address uncertainties around intervention design, optimal message frequency, and cost-effectiveness. | **Digital Literacy & Education-** Use of mobile phone voice messages tailored to specific populations and leveraging widespread mobile phone usage to increase healthcare access, particularly for non-literate women. |
| 24 | Otu et al., 2022 [42] | **Technology Infrastructure-** The integrated care package improved diagnostic services; **Community Engagement and Support-** Enhanced community relationships; **Support & training** (empowered nurses through task-shifting and demonstrated the feasibility of remote mHealth training.) | **Operational & Logistics Challenges-** Increased workload, organizational conflicts, and COVID-19 restrictions posed significant challenges to the smooth implementation of NCD care; **Network Coverage-** Technical issues with internet connectivity. | **Support & Training-** The study used mHealth training, remote supervision, and provision of necessary diagnostic equipment; **Community Engagement-** Community engagement incentives to enhance NCD care delivery. |
| 25 | Schmitz et al. 2022 [43] | **Improved Health Outcomes-** Increased Adherence to guidelines, referral rates, diagnosis and treatment, health care quality; **Community Engagement & Support; Support and Training-** Empowered nurses through task-shifting. | **Support & Training-** Training and Knowledge Gaps, Staff Turnover; **Operational & Logistical Challenges-** Contextual and Resource Constraints; **Health Outcomes-** Consultation Length; **Security Concerns; Network Coverage & Power Issues** | **Support & Training-** Training and Mentorship, Support for healthcare provider Motivation and Cognitive Relief; **Contextual & Resource Constraints-** Contextual Adaptation- tailored to the local context. |
| 26 | Shuaib et al. 2019 [44] | **Data Reporting & Quality**; **Technology Infrastructure**; **Support & Training**. **Cost & Financial Access-** Government ownership. | **Network Coverage and Power-** Lack of Internet. **Health Worker Support & Training-** Technical capacity, need for extensive training; **Contextual and Resource Constraints-** Limited state budgets. | **Community Engagement & Support-** Advocacy visits to state-level authorities, deployment of implementation officers (IOs); **Real-Time Data & Decision Making-** Phased implementation; **Support & Training-** Training programs; **Cost & Financial Access-** Financial support. |

*(Continued)*

**Table 3.** (Continued)

| S/No | First Author | Opportunities | Challenges | Strategies |
|---|---|---|---|---|
| 27 | Tegegne et al., 2018 [45] | **Real-Time Data & Decision Making-** Good geolocation and data for decision making); **Health outcomes** | **Operational & Logistical Challenges-** Limited supervision time for surveillance; **Contextual & Resource Constraints** (Difficulty in controlling variables); **Real-Time Data & Decision Making** (Dependence on real-time supervision) | **Support & Training-** Sensitization through supervision; **Contextual & Resource Constraints-** The study excluded health facilities that had received training within a year before the study; **Data Reporting & Quality-** The study disaggregated data to identify trends and improvements in AFP detection; **Real-Time Data & Decision Making-** Real-time feedback. |
| 28 | Tripathi et al. 2020 [46] | **Cultural & Gender Sensitivity-** Stigma and isolation, increased awareness of access; **Scalability & Population Reach** | **Cost & Financial Access-** Sustainability); **Cultural & Gender Sensitivity;** Language, Limited Awareness and Reach, Gender Inequality in Mobile Phone Ownership; **Network Coverage & Power-** Connectivity Issues; **Operational and Logistical Challenges-** Need for In-Person Follow-Up. | **Community Engagement & Support-** Community Support Integration; **Cultural & Gender Sensitivity-** Language and Cultural Sensitivity, Gender-Sensitive Approaches. |
| 29 | Uba et al. 2021 [47] | **Improved Data Reporting & Quality-** Increased Reporting Accuracy; **Improved Learning Opportunities-** Learning from Other Countries, Lesson Learning and Scale-Up; **Enhanced Technology infrastructure-** Networking and Coverage Solutions); **Support & Training-** Training | **Operational & Logistical Challenges-** Competing Activities in the State, Facilitator Limitations; **Support & Training-** Insufficient Hands-On Training; **Network Coverage & Power Issues Security Concerns**; **Support & Training-** Healthcare workers Staffing issues. | **Support & Training-** Follow-Up Mentorship, Separate Training Sessions for RI Tools; **Data Reporting & Quality-** Review of RI Dashboard; **Network Coverage & Power-** Identification of Better Internet Providers, Provision of Dedicated Internet Routers; **Cost & Financial Access-** Maximising Funding Opportunities. |
| 30 | Udenigwe et al. 2022 [48] | **Improved Cost & Financial Access-** Subsidized Healthcare Costs; **Increased Community Engagement & Support-** Access to Skilled Pregnancy Care; Increased Autonomy for Women; **Improved Health Outcomes** Enhanced Safety during pregnancy. | **Cultural & Gender Sensitivity-** Phone Ownership, Patriarchal Norms; **Community Engagement & Support-** Shared Use of Phones; **Cost & Financial Access** Ongoing Costs; **Support & Training-** Digital Literacy; **Network Coverage Issues** | **Community Engagement & Support-** Provision of free phones to pregnant women, Subsidized healthcare costs, Addressing social norms through community engagement; **Cultural & Gender Sensitivity-** Engaging male leaders (WDC chairpersons); **Technology Infrastructure-** Calls for infrastructure improvements in network coverage to better support the mHealth program. |
| 31 | Udenigwe et al. 2022 [49] | **Health Outcomes-** Positive Attitudes and Satisfaction, Perceived Effectiveness; **Community Engagement & Support**; **Cultural & Gender Sensitivity-** Increased Health Awareness, Subsidized Phones; **Support & Training-** Competence and Training. | **Contextual & Resource Constraints-** Limited Resources; **Cultural & Gender Sensitivity-** Gender Inequality, Cultural Barriers, and Social norms created suspicion when women interacted with male WDC chairpersons or drivers, especially at night; **Cost & Financial Access-** Lack of financial investment from local governments and community; **Health Worker Support & Training-** (Inadequate Workforce. | **Community Engagement & Support-** Community Participation by engaging WDC chairpersons, Subsidization of Phones; **Support & Training-** Training and Education; **Cost & Financial Access-** Recommendations for Sustainability; **Cultural & Gender Sensitivity-** Cultural Sensitivity. |

and digital health tools efficiently reached large populations [26,27,31,32,41,42,44,46,47,55]. These tools significantly enhanced health education among non-literate groups [40,42,44,48].

**Frequency of occurrences of themes across reviewed studies**

Table 4 presents the distribution of key themes across the reviewed studies, categorized into opportunities, challenges, and strategies. The most frequently cited themes were community engagement and support, health outcomes, and support and training, each appearing consistently across all three categories. Technology infrastructure was also prominent, particularly as an opportunity and a challenge.

Less frequently discussed but nonetheless important themes included cultural and gender sensitivity, digital literacy and education, and real-time data and decision-making. Themes such as cost and financial access, operational and logistical issues, network coverage and power, and security concerns were mainly reported as challenges, with fewer associated strategies. Notably, operational issues and network constraints were not identified as opportunities in any study, underscoring their persistent and unresolved nature.

## Discussion

This review examined the integration of DHTs in Nigeria's healthcare system, highlighting both challenges and opportunities. Though DHTs can enhance health outcomes and access, we observed obstacles facing their implementation, such as infrastructural limitations, lack of unified electronic health records, limited geographic spread, insufficient implementation time, and issues with continuity and scalability.

The integration of digital health technologies in Nigerian healthcare settings, as explored in this scoping review, offers significant insights through the lens of the study's conceptual framework, in line with the Nigeria Digital in Health Initiative (NDHI), which aims to improve healthcare delivery by integrating digital technologies, such as telemedicine and mobile health (mHealth), into the healthcare system. This initiative encourages collaboration between the government, private sector, and international partners to address infrastructural and operational challenges [57].

The study's findings reveal promising opportunities and persistent challenges associated with digital health interventions, presenting a comprehensive picture of their impact on healthcare delivery. The review also identified several strategies central to optimizing digital health interventions. It presents thorough synthesis of themes across multiple studies, particularly the emphasis on support and training, community engagement, sustainability and technology infrastructure. These themes align well with the Digital Health Ecosystem framework [14], which posits that successful digital health integration requires a synergistic approach involving multiple health system components. The frequency of the support and

**Table 4. Frequency of themes across reviewed studies.**

| Theme | Opportunities | Challenges | Strategies |
|---|---|---|---|
| Community Engagement & Support | 19, 21, 24, 25, 30, 31, 32, 35, 42, 48 | 24 | 19, 20, 25, 30, 42, 48, 49 |
| Health Outcomes | 19, 20, 21, 24, 32, 33, 38, 49 | 20, 21, 29, 43 | 31, 35, 47 |
| Support & Training | 22, 27, 28, 30, 42, 44 | 27, 29, 43 | 22, 28, 30, 42, 47 |
| Technology Infrastructure | 21, 26, 27, 33, 39, 40 | 29, 34, 42 | 34, 39 |
| Cultural & Gender Sensitivity | 46, 48, 49 | 38, 46, 48 | 46, 48, 49 |
| Digital Literacy & Education | 24, 33, 35, 41 | 23, 24, 33 | 33, 41 |
| Real-Time Data & Decision-Making | 22, 27 | 45 | 27, 45 |
| Cost & Financial Access | 24, 33, 48 | 22, 30, 34 | 24, 33, 48 |
| Operational & Logistical Issues | – | 29, 34, 42, 43 | |
| Network Coverage & Power | – | 29, 34, 43 | – |
| Security Concerns | – | 30, 32, 43 | 32 |

training theme across reviewed papers underscores its critical role in the digital health ecosystem. However, challenges such as limited resources and varying levels of technological literacy across regions can hinder the implementation of robust training programs. Opportunities exist to leverage partnerships with educational institutions and technology providers to overcome these barriers and ensure that training is accessible and practical.

By enhancing the competence and motivation of healthcare workers through continuous training, digital health technologies can significantly improve service delivery and patient outcomes. This strategy addresses the opportunity to enhance the effectiveness of digital health tools by ensuring that users are well-trained and confident in their use. This finding also aligns with the WHO model's emphasis on strengthening health workforce capabilities, vital for sustaining health system improvements [15].

Community engagement (such as the involvement of local leaders, healthcare workers and patients in the planning and executing digital health initiatives) also emerged as a pivotal factor, reflecting the WHO model's focus on strengthening community participation and ownership of health initiatives [15]. This community-centric approach fosters trust and ensures that digital health solutions are culturally sensitive and tailored to the population's needs. Nonetheless, resistance to new technologies and varying levels of community readiness can impede progress. Addressing these challenges involves implementing strategies like participatory design [58] and localized awareness campaigns that can bridge the gap between new technologies and community needs. The study identifies successful community engagement strategies, such as localized SMS messaging and digital education programs, which indicate how digital health interventions can be effectively implemented in diverse settings.

Technology infrastructure, another prominent theme, highlights the importance of a robust infrastructural foundation for the success of health interventions. The findings reveal that significant challenges remain while opportunities are related to increasing mobile and internet penetration in Nigeria. Issues such as poor network coverage, epileptic power supply, and inadequate technological infrastructure continue to hinder the full potential of digital health solutions. Strategic investments and partnerships with telecom companies and local businesses can help address these issues, creating a more reliable and accessible technological environment essential to ensuring that digital health technologies can be reliably deployed and sustained.

The review process brought to light emerging but critical aspects of digital health integration, including real-time data reporting and cultural sensitivity, which are less frequently discussed but vital for the long-term success of digital health interventions, particularly in low- and middle-income countries (LMICs) such as Nigeria. The limited focus on these areas presents an opportunity for further research and a call for greater emphasis on future initiatives. Real-time data reporting, for example, is indispensable in making informed decisions and enabling timely health interventions [59], yet it remains underdeveloped in many healthcare systems. Challenges such as inadequate data management and the absence of robust systems —including information management platforms, data governance frameworks, technological infrastructure, monitoring and evaluation tools, decision support systems, and integrated systems— pose significant barriers to fully capitalizing on this potential. Investments in advanced data management technologies and training healthcare personnel in efficient data handling are essential to harness the full power of real-time reporting, enabling quicker responses to health emergencies and improving patient outcomes.

Cultural and gender sensitivity emerged as a theme in ensuring equitable access and effectiveness of digital health interventions. Adopting these technologies requires addressing local customs and gender dynamics to avoid alienating target populations. This entails recognizing gender-specific roles and healthcare access disparities, engaging influential community members like elders or religious leaders, respecting traditional health beliefs, promoting language inclusivity, and ensuring culturally sensitive approaches to privacy and data sharing.

However, significant challenges persist, including entrenched cultural norms and gender disparities, which often impede the scalability and acceptance of digital health solutions. Overcoming these barriers involves tailoring interventions to reflect local contexts while addressing gender inequities, enhancing their relevance, inclusivity, and overall impact. By

aligning these technologies with community-specific needs, digital health solutions can become more scalable and effective in diverse cultural settings.

The study's identification of other challenges, including operational and logistical issues, scalability, and financial barriers, reflect significant obstacles to successfully implementing digital health interventions. The findings suggest that while digital health technologies promise to improve healthcare access and outcomes, addressing these challenges is crucial for widespread adoption and long-term success. A significant issue identified in the review was the one-off nature of many DHT integration studies, which lacked continuity and sustained impact. This short-term approach limited the long-term benefits and scalability of digital health interventions. Furthermore, the limited geographic spread of many of these studies further hampered the widespread adoption of DHTs.

A key strategy for overcoming challenges in integrating digital health in Nigeria is implementing a standardized EHR system across all healthcare institutions. This unified system would facilitate the seamless transfer of patient information, eliminating the need for patients to repeatedly provide their medical history when visiting healthcare facilities in different regions. Such continuity of care is vital for enhancing health outcomes and ensuring that digital health technologies foster a more efficient, interconnected healthcare system nationwide.

While this review provides valuable insights into digital health integration from the perspectives of healthcare providers, patients, and communities, an important gap lies in the limited representation of voices from local technology developers, implementers, and key government agencies. These stakeholders play a central role in shaping the feasibility, scalability, and sustainability of digital health initiatives. Their firsthand experiences with policy barriers, infrastructural constraints, and deployment challenges are essential for understanding the full landscape of DHT integration in Nigeria.

### Key findings

The study highlights significant improvements in health outcomes associated with digital health tools, demonstrating their potential to create more equitable, efficient, and responsive health systems.

Opportunities were particularly evident in the potential for scaling these technologies to reduce health disparities, enhance disease surveillance, expand healthcare access, and streamline service delivery. Key enablers include advancements in real-time data systems, effective community engagement strategies, and improvements in infrastructure. These factors collectively support progress in areas such as maternal health, chronic disease management, and emergency response.

The review also highlights persistent challenges that hinder the full realization of digital health technologies in Nigeria. These include inadequate infrastructure, low digital literacy, unreliable network coverage, financial constraints, and socio-cultural barriers that affect equitable access. Addressing these issues will require integrated strategies that combine technological innovation with supportive policy and community-driven approaches. Until these systemic gaps are effectively resolved, the scalability and sustained impact of many digital health interventions may remain limited.

### Limitations of the study

While inclusive, the broad approach of the scoping review may limit the depth of analysis for specific digital health interventions. Variations in study design and regional focus among the included studies could affect the generalizability of the findings. The rapid evolution of digital health technologies means that the findings represent a snapshot in time, potentially necessitating further targeted studies to capture ongoing advancements and emerging trends.

Additionally, the inclusion criteria were guided by the scope of empirical literature available at the time of the review, which largely centered on user-level experiences and outcomes. As a result, perspectives from system-level stakeholders such as digital health technology developers, private implementers, and government agencies were underrepresented, limiting insights into the broader structural and policy environment surrounding DHT implementation.

Future research should be expanded to include the experiences and insights of local digital health technology developers, private-sector implementers, and relevant government institutions responsible for health and ICT policy. Including these key stakeholders will provide a more holistic view of the digital health ecosystem, capturing implementation realities, policy environments, and technological readiness. Their inclusion is crucial for developing context-appropriate strategies, ensuring better alignment between innovation, infrastructure, and governance, and ultimately achieving sustainable digital health integration in Nigeria.

## Conclusion

This scoping review has highlighted challenges, opportunities and strategies associated with integrating DHTs within the Nigerian healthcare system. While DHTs presented significant potential for enhancing health outcomes, improving access to care, and streamlining healthcare delivery, their implementation remained fraught with obstacles. Key challenges include infrastructural limitations, lack of unified EHR systems, and restrictions on the continuity and scalability of several digital health interventions. Furthermore, the limited geographic spread of several DHT projects and the absence of sustained, long-term studies were also identified. However, emerging themes like real-time data reporting, cultural sensitivity, and gender considerations offer new opportunities for more inclusive and effective digital health solutions. Addressing these issues while building on the strengths of existing systems can transform healthcare delivery and ensure that the benefits of DHTs are fully realized.

## Supporting information

**S1 Appendix: Preferred Reporting Items for Systematic reviews and Meta-Analyses extension for Scoping Reviews (PRISMA-ScR) checklist.**
(DOCX)

**S2 Appendix: PubMed search strategy - web of science search strategy - CINAHL Search Strategy - SCOPUS Search Strategy.**
(DOCX)

**S3 Appendix: List of excluded studies and reasons for exclusion.**
(DOCX)

**S4 Appendix: Data extraction sheet.**
(DOCX)

**S5 Appendix: Summary of the mixed method appraisal tool for quality assessment of studies included.**
(XLSX)

## Author contributions

**Conceptualization:** Adaeze E. Egwudo, Ayodapo Oluwadare Jegede, Tolulope A. Oyeniyi, Nkolika J. Ezekwelu, Samirah N. Abdu-Aguye, Olunike R. Abodunrin, Folahanmi T. Akinsolu, Olajide O. Sobande.

**Data curation:** Adaeze E. Egwudo, Ayodapo Oluwadare Jegede, Tolulope A. Oyeniyi, Nkolika J. Ezekwelu, Samirah N. Abdu-Aguye, Olunike R. Abodunrin, Folahanmi T. Akinsolu.

**Formal analysis:** Adaeze E. Egwudo, Ayodapo Oluwadare Jegede, Tolulope A. Oyeniyi, Nkolika J. Ezekwelu, Samirah N. Abdu-Aguye.

**Funding acquisition:** Folahanmi T. Akinsolu, Olajide O. Sobande.

**Investigation:** Adaeze E. Egwudo, Ayodapo Oluwadare Jegede, Nkolika J. Ezekwelu, Samirah N. Abdu-Aguye.

**Methodology:** Adaeze E. Egwudo, Ayodapo Oluwadare Jegede, Tolulope A. Oyeniyi, Nkolika J. Ezekwelu, Samirah N. Abdu-Aguye, Folahanmi T. Akinsolu, Olajide O. Sobande.

**Project administration:** Ayodapo Oluwadare Jegede, Tolulope A. Oyeniyi, Oluchukwu P. Okeke, Olunike R. Abodunrin, George U. Eleje, Folahanmi T. Akinsolu, Olajide O. Sobande.

**Resources:** Folahanmi T. Akinsolu, Olajide O. Sobande.

**Software:** Ayodapo Oluwadare Jegede, Oluchukwu P. Okeke, Olunike R. Abodunrin, Folahanmi T. Akinsolu.

**Supervision:** Olunike R. Abodunrin, George U. Eleje, Folahanmi T. Akinsolu, Olajide O. Sobande.

**Validation:** Adaeze E. Egwudo, Tolulope A. Oyeniyi, Nkolika J. Ezekwelu, Samirah N. Abdu-Aguye, Azuka P. Okwuraiwe, Olunike R. Abodunrin.

**Visualization:** Adaeze E. Egwudo, Ayodapo Oluwadare Jegede, Tolulope A. Oyeniyi, Nkolika J. Ezekwelu, Samirah N. Abdu-Aguye, Chizaram A. Onyeaghala, Theresa O. Ozoude, Muritala O. Suleiman, Grace O. Aziken, George U. Eleje, Folahanmi T. Akinsolu, Olajide O. Sobande.

**Writing – original draft:** Adaeze E. Egwudo, Ayodapo Oluwadare Jegede, Tolulope A. Oyeniyi, Nkolika J. Ezekwelu, Samirah N. Abdu-Aguye.

**Writing – review & editing:** Adaeze E. Egwudo, Ayodapo Oluwadare Jegede, Tolulope A. Oyeniyi, Nkolika J. Ezekwelu, Samirah N. Abdu-Aguye, Azuka P. Okwuraiwe, Chizaram A. Onyeaghala, Theresa O. Ozoude, Muritala O. Suleiman, Grace O. Aziken, Oluchukwu P. Okeke, Olunike R. Abodunrin, George U. Eleje, Folahanmi T. Akinsolu, Olajide O. Sobande.

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
