## [Decision Letter · Decision Letter 0]

Response to Reviewers
Revised Manuscript with Track Changes
Manuscript
**Journal Requirements:**
**Additional Editor Comments (if provided):**
**Reviewers' Comments:**

**Comments to the Author**

1. Does this manuscript meet PLOS Digital Health’s publication criteria?

Reviewer #1: Yes

Reviewer #2: Yes

2. Has the statistical analysis been performed appropriately and rigorously?

Reviewer #1: Yes

Reviewer #2: Yes

3. Have the authors made all data underlying the findings in their manuscript fully available (please refer to the Data Availability Statement at the start of the manuscript PDF file)?

Reviewer #1: Yes

Reviewer #2: Yes

4. Is the manuscript presented in an intelligible fashion and written in standard English?

Reviewer #1: Yes

Reviewer #2: Yes

Reviewer #1: Overview:

This is a very well-executed and interesting scoping study. As an entry point into this literature, it is a valuable addition to public health scholarship, and merits wide circulation.

The paper is appropriate for publication as-is. Below I make some suggestions that might help with clarity for future readers.

(line number: comment)

41: This sentence implies that the number of selected studies was N, of which a subset, 31, demonstrated improved health outcomes. But the total number of included studies was 31, and improved health outcomes was not a criterion. So more accurate might be to remove line 41, and to say at line 39 "A total of 31 observational and experimental studies were included".

85: typo "holds" -> "hold"

145 - 146: Perhaps some of these items, selected by the authors according to the value of the perspectives they offer on the topic, could be included in the references. This paper serves as a valuable gateway to the literature for those working on DHT. Including select references is another aspect of this gateway role.

270 - 271: This sentence is puzzling since it refers to SMS interventions, which require literacy. Does "non-literate" refer to health information rather than reading ability? If so, please clarify.

Table 3:

1. The outer parentheses could be removed from all entries.

2. Adding periods (punctuation) to all entries will help the reader know whether boxes at the bottom of a page are complete or continued on the next page.

3. Because of the large amount of text, perhaps breaking this table up into three tables ("Opportunities", "Challenges", "Strategies") would make the contents more accessible for the reader (I don't know).

276 - 304: (Section "Frequency of Occurrences of Themes Across Reviewed Studies")

1. There is some mismatch between adjectives and frequencies. Egs:

288: "pressing need" vs only 3 papers that mentioned challenges;

298: "significant" vs the count of papers that raised the issue;

291: "less frequently discussed" vs "crucial" in line 372.

2. Perhaps there are two metrics at work here: simple frequency (number of papers) and weighted importance (which can be determined by the reviewers). If so, clarifying this would be useful.

2. Perhaps this section can be more compactly captured in a table for better readability.

The following items are all on the same theme: The paper does not need to claim its own strength - the reader can readily deduce this.

321: Rephrase, eg to "It presents thorough synthesis ..."

346 - 347: Rephrase, eg to "The study identifies successful ... programs, which indicate how digital..."

358: Maybe rephrase, eg to "The review process brought to light emerging but critical..."

401: Rephrase, eg to "Key findings". Then include in this paragraph findings as to key challenges/roadblocks. Or line 401 could be removed, and lines 402 - 409 could be folded into "Conclusions"

389 - 390: This is super and important and interesting to funders and groups planning trials. Can details be included in the "Frequency of Occurrences of Themes Across Reviewed Studies" section, to direct the reader to relevant papers?

416 - 417: I would argue that this is not a limitation of the study, but rather a key challenge highlighted by the study. It thus belongs in the previous paragraph ("Key findings").

Thank you for an interesting and well-written paper.

Reviewer #2: The study provides appreciably comprehensive insights into understanding the challenges and opportunities of integrating digital health technologies into the healthcare system in Nigeria, examining a myriad of existent publications, as well as employing observational and experimental studies among major stakeholders.

The use of ‘Observational and experimental studies involving healthcare workers, patients, caregivers, or other stakeholders impacted by DHT integration’ is an important approach in examining the dynamics among these stated stakeholders. Studies, specifically conducted within Nigeria are more important in offsetting ambiguities given that health system dynamics do differ.

However, there is a need to include actual local (within Nigeria) Digital Health Technology service/product producers and implementers in the studies to gain first hand insight into opportunities they are seeing and challenges they have faced on-ground. This being first hand producers and implementers of the digital health services/products, will provide first-hand knowledge in as far as challenges and opportunities therein are concerned. Inclusion of the stated government agencies, would help to understand the status of policies and infrastructure that may favour or hinder the implementation of digital health technologies.

In line “99 1:9083 [12], far below the WHO's recommended ratio of 1:600 [ref]”, there is need a reference.

**Do you want your identity to be public for this peer review?** For information about this choice, including consent withdrawal, please see our Privacy Policy

Reviewer #1: No

Reviewer #2: No

**Figure resubmission:****Reproducibility:** To enhance the reproducibility of your results, we recommend that authors of applicable studies deposit laboratory protocols in protocols.io, where a protocol can be assigned its own identifier (DOI) such that it can be cited independently in the future. Additionally, PLOS ONE offers an option to publish peer-reviewed clinical study protocols. Read more information on sharing protocols at https://plos.org/protocols?utm_medium=editorial-email&utm_source=authorletters&utm_campaign=protocols

---

## [Editor Report · Decision Letter 1]

Integrating digital health technologies into the healthcare system: challenges and opportunities in Nigeria

PDIG-D-24-00577R1

Dear Dr Jegede,

We are pleased to inform you that your manuscript 'Integrating digital health technologies into the healthcare system: challenges and opportunities in Nigeria' has been provisionally accepted for publication in PLOS Digital Health.

Best regards,

Calvin Or, PhD

Section Editor

PLOS Digital Health